EMBO
Molecular Medicine

# Somatically mutated ABL1 is an actionable and essential NSCLC survival gene

Ewelina Testoni[1], Natalie L Stephenson[1,†], Pedro Torres-Ayuso[1,†], Anna A Marusiak[1], Eleanor W Trotter[1], Andrew Hudson[1], Cassandra L Hodgkinson[2], Christopher J Morrow[2], Caroline Dive[2] & John Brognard[1,*]

## Abstract

The lack of actionable mutations in patients with non-small cell lung cancer (NSCLC) presents a significant hurdle in the design of targeted therapies for this disease. Here, we identify somatically mutated *ABL1* as a genetic dependency that is required to maintain NSCLC cell survival. We demonstrate that NSCLC cells with *ABL1* mutations are sensitive to ABL inhibitors and we verify that the drug-induced effects on cell viability are specific to pharmacological inhibition of the ABL1 kinase. Furthermore, we confirm that imatinib suppresses lung tumor growth *in vivo*, specifically in lung cancer cells harboring a gain-of-function (GOF) mutation in ABL1. Consistent with structural modeling, we demonstrate that mutations in *ABL1* identified in primary NSCLC tumors and a lung cancer cell line increase downstream pathway activation compared to wild-type ABL1. Finally, we observe that the ABL1 cancer mutants display an increased cytosolic localization, which is associated with the oncogenic properties of the ABL1 kinase. In summary, our results suggest that NSCLC patients with *ABL1* mutations could be stratified for treatment with imatinib in combination with other therapies.

**Keywords** ABL1 mutations; dasatinib; imatinib; non-small cell lung cancer
**Subject Categories** Cancer; Respiratory System

## Introduction

Non-small cell lung cancer (NSCLC) accounts for approximately 80% of lung cancers and the majority of these patients have metastatic disease at diagnosis with < 5% surviving for more than 5 years (Politi & Herbst, 2015). In Western populations, only approximately 20% of patients with lung adenocarcinoma can be stratified for treatment with targeted therapies aimed at mutationally activated kinases such as EGFR or EML4-ALK (Heist and Engelman, 2012). Therefore, it is necessary to elucidate novel genetic drivers that are essential to maintain lung cancer cell survival.

Accumulating cancer genomic data reveal mutations in pharmacologically tractable targets at an ever-increasing rate. However, time-consuming functional studies must be performed to assess the consequences of the mutations and determine whether the mutation represents a genetic dependency that is required to maintain cancer cell viability (Brognard *et al*, 2011; Fawdar *et al*, 2013). Genetic dependencies are defined as somatically mutated enzymes that promote cancer cell survival in the context of other complex genetic alterations, but may not necessarily be initiating driver events. ABL1 and ABL2 display mutations in approximately 1.5% and 4% of NSCLC cases, respectively, and ABL2 is amplified in approximately 8% of NSCLC cases (Appendix Fig S1A and B, adapted from cBio Cancer Genomics Portal). Additionally, ABL1 and ABL2 are mutated at a frequency of approximately 3% and 10% in SCLC, respectively (Appendix Fig S1A and B).

The BCR-ABL oncoprotein is the primary driver of chronic myelogenous leukemia (CML) (Hunter, 2007; Druker, 2008). The characteristic feature of CML is the presence of the Philadelphia chromosome, the product of the reciprocal translocation between chromosomes 9 and 22 (Hunter, 2007; Druker, 2008). The product of this translocation is the BCR-ABL fusion protein, where the ABL tyrosine kinase domain is constitutively active (Hunter, 2007). Inhibition of the kinase activity by small molecule inhibitors, such as imatinib, results in robust tumor responses in patients with CML carrying this translocation (Druker *et al*, 1996; Druker, 2008). This crucial discovery has helped usher in an era of personalized precision medicine aimed at inhibiting mutationally activated enzymes, such as kinases, that are required to maintain cancer cell proliferation and survival. However, in solid tumors the role of the ABL kinases in tumorigenesis is relatively unexplored. ABL1 has been implicated to be activated and promote phenotypes associated with cancer, by mechanisms including loss of the ABL1 negative regulator FUS1 (Lin *et al*, 2007). ABL kinases regulate a range of cellular processes including cell survival, proliferation, motility, cell polarity, and oxidative stress responses, suggesting deregulation of the kinases may promote tumorigenic phenotypes (Greuber *et al*, 2013). However, the exact

1  Signalling Networks in Cancer Group, Cancer Research UK Manchester Institute, The University of Manchester, Manchester, UK
2  Clinical and Experimental Pharmacology Group, Cancer Research UK Manchester Institute, The University of Manchester, Manchester,UK
  *Corresponding author. Tel: +44 161 306 5301; Fax: +44 161 918 7134; E-mail: john.brognard@cruk.manchester.ac.uk
  †These authors contributed equally to this work

 

role of genetically altered ABL kinases in solid tumors remains largely unknown (Greuber *et al*, 2013). Depending on cellular context and cancer type, the proliferative effects of ABL1 can be exerted by controlling upregulation of c-myc or cyclin D1 (Furstoss *et al*, 2002; Kundu *et al*, 2015). In addition, in breast cancer, ABL1 has been reported to phosphorylate the sliding clamp protein, PCNA at Tyr211 leading to association with chromatin and increased cell proliferation (Zhao *et al*, 2014). Lastly, both ABL1 and ABL2 are essential for the invasive phenotype of melanoma cells, where ABL kinases were shown to directly regulate expression and activation of matrix metalloproteinases (Ganguly *et al*, 2012). Combined, this highlights that deregulation of the ABL kinases contributes to acquisition of tumorigenic phenotypes in solid tumors and provides the rationale for investigating the functional consequences of somatic mutations observed in the ABL kinases in lung cancer.

Here, we demonstrate that mutations in ABL1, but not ABL2, represent an actionable genetic dependency that is required to maintain NSCLC cell survival. These results could be rapidly translated to the clinic as lung cancer patients with *ABL1* mutations could be stratified for treatment with clinically approved ABL inhibitors, such as imatinib, in combination with other therapies.

## Results

### Lung cancer cells with ABL1 mutations are sensitive to imatinib and dasatinib

In an effort to identify novel genetic dependencies of lung cancer, we searched the CCLE (Cancer Cell Line Encyclopedia) database for NSCLC cell lines that harbored mutations in ABL1 or ABL2 (Barretina *et al*, 2012) and identified four lung adenocarcinoma cell lines H1915 (ABL1-R351W), H2110 (ABL1-G340L), H650 (ABL2-W469C), and H1623 (ABL2-Y399C). Mutational analysis was performed using web applications to predict the functional impact of the ABL1/2 mutations in these lung cancer cell lines (Appendix Table S1). All mutations occurred in the kinase domain and mutations in ABL1 identified in lung cancer cell lines and primary tumor tissues are depicted in Fig EV1. To determine whether somatically mutated ABL1/2 may be required to maintain lung cancer cell survival, we treated this panel of NSCLC cancer cell lines harboring mutations in either ABL1 or ABL2 (H1915, H2110, H650, and H1623), with ABL inhibitors, and monitored cell viability. We used negative control cell lines, which included a normal immortalized human bronchial epithelial cell line (Beas-2B), a cell line expressing constitutively active EGFR (HCC827) and a lung cancer cell line with no known drivers (H2126; mutations and copy number alterations for all cancer cell lines are included in Dataset EV1). We also used the A549 cell line; these cells lack expression of the FUS1 protein, a negative regulator of ABL1 (Lin *et al*, 2007). To determine whether long-term treatment with imatinib would affect colony formation in this panel of cell lines, the cells were treated with 3.7 μM imatinib for two weeks. Imatinib is a very specific ABL inhibitor, which binds the DFG-out inactive conformation of the ABL kinases and locks these kinases in an inactive state (Schindler *et al*, 2000). This concentration was used based on a dose–response curve using H1915, H2110, A549, and H650 cell lines (Fig EV2A). A significant reduction in colony formation was observed in both the H1915

(homozygous for R351W mutation) and H2110 (heterozygous for the G340L mutation) lung cancer cell lines, whereas the control cell lines and cell lines carrying ABL2 mutations (H650 and H1623) displayed no change in colony formation when treated with imatinib (Fig 1A). Based on previous reports, we utilized CRKL phosphorylation to determine endogenous ABL activity in cells (Druker, 2008; Shah *et al*, 2008; Seo *et al*, 2010). Imatinib inhibited phosphorylation of CRKL to varying degrees in all cell lines, with more conspicuous inhibition observed in the Beas-2B, H1915, H2110, and H650 cell lines (Fig 1B). To validate these results, we performed an MTT assay across the panel of lung cancer cell lines and observed that the H1915 and H2110 lung cancer cells were the most sensitive to imatinib (Appendix Fig S2A). This observation is consistent with the colony formation assay results and indicates lung cancer cells with ABL1 mutations are sensitive to imatinib.

To further explore the possibility that NSCLC cells with ABL1 mutations are sensitive to ABL inhibitors, we treated this panel of lung cancer cell lines with 40 nM dasatinib (a clinically relevant concentration) that was selected based on dose curves in the H1915, H2110, H650, H1623, and A549 cell lines (Fig EV2B). Dasatinib is a class I ATP competitive inhibitor that targets the active kinase conformation of several tyrosine kinases including ABL1/2 (Tokarski *et al*, 2006). We observed a significant reduction in cell viability in cells harboring ABL1 mutations or the A549 cells that lack FUS1, but not in cells with ABL2 mutations or WT ABL1/2 (Fig 1C). Dasatinib decreased CRKL phosphorylation in all cell lines except the H1623 cell line, suggesting dasatinib is targeting the ABL kinases in these cells (Fig 1D). To determine whether this concentration also inhibits the related kinase SRC, phosphorylation of SRC and SRC's direct downstream substrate FAK was assessed in all cell lines (Fig 1D). We observed variable effects on SRC activity in response to dasatinib treatment across the cell line panel, with no significant reduction in SRC activity in H1915 and H2110 cells, suggesting that SRC inhibition does not significantly contribute to dasatinib-induced cell toxicity in these two cell lines. In addition, we treated a panel of these cells (H1915, H650, H1623, and A549) with dasatinib and monitored colony formation (Appendix Fig S2B). Consistent with MTT results, treatment of the H1915 and A549 cells with 40 nM dasatinib resulted in a significant reduction in colony formation, with no effect on colony formation in cells with ABL2 mutations (Appendix Fig S2B). In general, these results are consistent with the responses observed when cell lines were treated with imatinib, except for A549 cells, in which imatinib did not reduce cell viability or colony-forming potential (Fig 1A and Appendix Fig S2A). Therefore, dasatinib-mediated toxicity observed for A549 cells is likely attributed to off-target effects of this drug.

To assess whether sequential treatment with ABL inhibitors would further diminish the cell growth, we performed a replating experiment. Here, we replated cells exposed to imatinib and treated the cells with a second round of the drug. For both rounds of imatinib treatment, the drug concentration was reduced to 2 μM to account for excessive toxicity normally observed at 4 μM and to allow sufficient cell density during the replating procedure. We did observe a further reduction in colony formation, but the response was still not complete, indicating other drivers are contributing to cell survival (Appendix Fig S3A). Additionally, in subsequent imatinib treatments, the cells often did not survive the replating, indicating ABL1 is required for the colony-forming potential of the

**A**

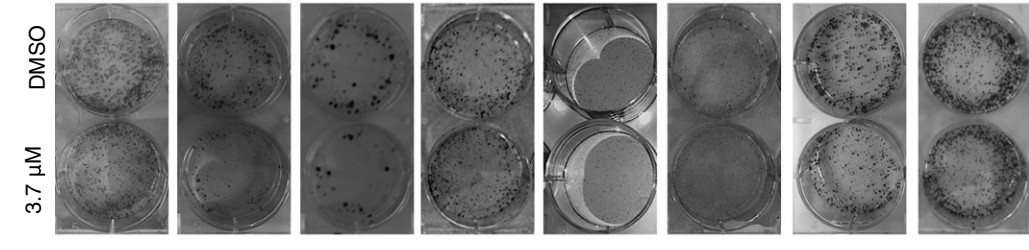

**B**

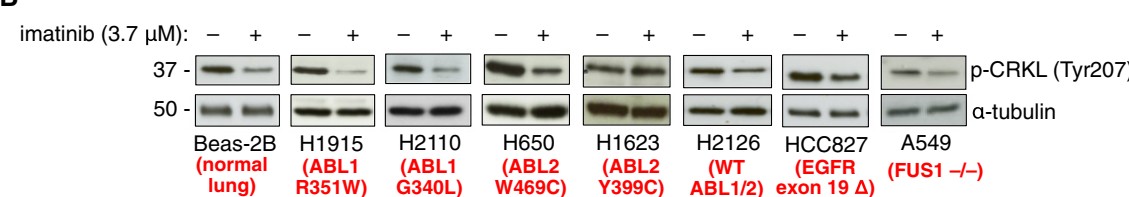

**C**

**D**

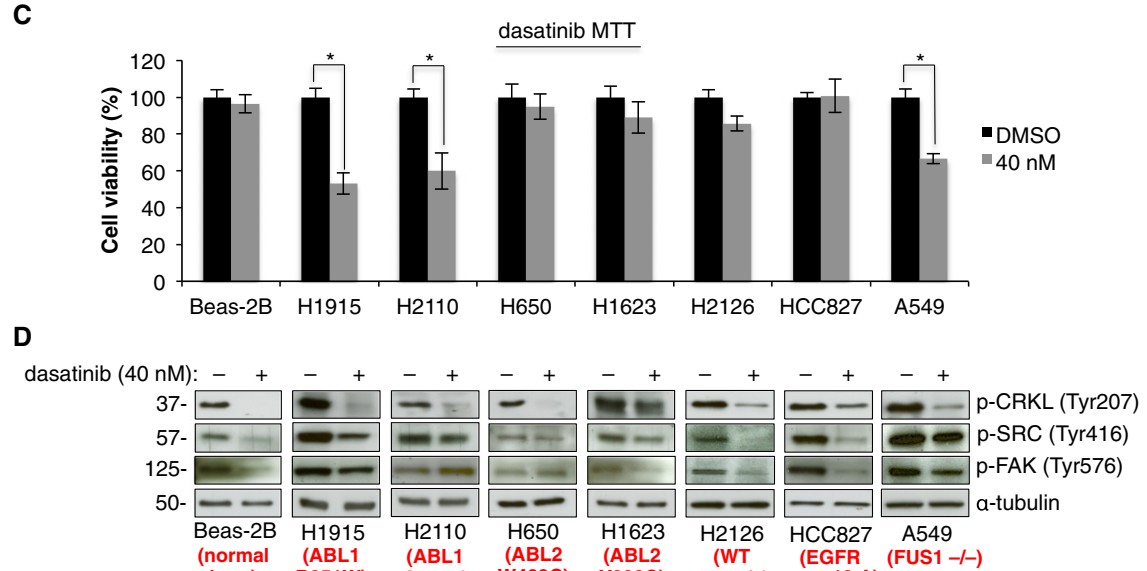

**Figure 1. NSCLC with ABL1 mutations are sensitive to ABL inhibitors.**

A   Colony formation assay (CFA) with imatinib with representative images of colonies formed by each cell line within the panel. Error bars represent mean ± S.E.M., $n = 9$, *$P < 0.001$ by two-tailed unpaired Student's $t$-test (H1915 $P = 0.0004$, H2110 $P = 0.0001$).

B   Western blot analysis after 1 h treatment with imatinib.

C   MTT assay for NSCLC cell lines and Beas-2B cells with dasatinib or DMSO. Error bars represent mean ± S.E.M., $n = 9$, *$P < 0.001$ by two-tailed unpaired Student's $t$-test (H1915 $P = 1.09 \times 10^{-5}$, H2110 $P = 0.009$ and A549 $P = 6 \times 10^{-6}$).

D   Western blot analysis after 1 h treatment with dasatinib.

Source data are available online for this figure.

cancer cells. This further emphasizes the dependence of H1915 cells on ABL1 to promote tumorigenic phenotypes and demonstrates an increased sensitivity to imatinib (Appendix Fig S3A). In addition, we did not observe any additional mutations, other than R351W, acquired in ABL1 in cells resistant to imatinib (Appendix Fig S3B).

To determine whether lung cancer cells containing ABL2 amplifications are sensitive to ABL inhibitors, we utilized two lung adeno-carcinoma cell lines (H810 and H1355) and a small cell lung cancer cell line (H1048), all harboring ABL2 amplifications, and subjected to treatment with imatinib. Expression of ABL2 was increased in H810 and H1048 cells, but there was no relative increase observed in H1355 cells, compared to other lung adenocarcinoma cell lines that lacked ABL2 amplification (Fig EV2C). Treatment with imatinib moderately suppressed colony formation, only in the H810 cells, suggesting ABL2 amplification will not be a potent driver in lung cancer (Fig EV2D).

### Drug-resistant ABL1 rescues dasatinib- and imatinib-induced cytotoxicity

To determine whether the cell toxicity observed in lung cancer cells harboring ABL1 mutations can be attributed to inhibition of this kinase, we treated H1915 cells with ABL inhibitors and re-expressed the drug-resistant ABL1-T315I mutant (Bixby & Talpaz, 2009). We used the H1915 cell line because it is amenable to transient transfection and generation of tetracycline-inducible stable cell lines. Transient overexpression of ABL1-T315I mutant promoted resistance to dasatinib or imatinib as assessed by CRKL phosphory-lation, which was maintained in inhibitor-treated cells expressing the drug-resistant mutant (Fig 2A). Consistent with transient over-expression, tetracycline-induced overexpression of the ABL1-T315I in H1915 cells maintains CRKL phosphorylation in the presence of dasatinib or imatinib (Fig 2B). Furthermore, the reduction in cell viability in H1915 cells treated with either dasatinib or imatinib was partially or fully rescued by expression of drug-resistant ABL1-T315I (Fig 2C). A reduced concentration of dasatinib was used due to the partial inhibition of ABL1-T315I (Fig 2A–C). These results demon-strate that the drug-induced decrease in cell viability observed in H1915 lung cancer cells, homozygous for the R351W mutation, is due to inhibition of somatically mutated ABL1, which represents a druggable genetic dependency in this cell line.

### Depletion of somatically mutated ABL1 reduces viability of lung cancer cells

To further verify our inhibitor studies, we depleted ABL1 from the Beas-2B, H1915, and H2110 cells using three unique siRNAs. In all cell lines, depletion of ABL1 resulted in a significant reduction in the ABL1 protein level and decreased phosphorylation of the downstream substrate CRKL (Fig 2D–F). Consistent with our inhibitor data, deple-tion of ABL1 from both lung cancer cell lines harboring ABL1 muta-tions resulted in a significant reduction in cell viability compared with Beas-2B cells (Fig 2D–F). In addition, there was no decrease in cytotoxicity detected in additional cell lines depleted of the ABL1 kinase (A549 and H2126), consistent with results from our studies with imatinib (Fig EV3A). In addition, depletion of ABL2 from lung cancer cells lines with ABL2 amplifications did not result in a compel-ling reduction in viability (Fig EV3B), although a consistent minor

reduction in viability was observed in the H810 cells, suggesting ABL2 contributes to promoting cell viability in this cell line. In summary, our data further confirms that imatinib- and dasatinib-induced cell toxicity is due to specifically targeting the ABL1 kinase, rather than inhibition of other kinases targeted by these inhibitors.

### Imatinib reduces tumor growth *in vivo*

To determine the activity of imatinib against NSCLC cells harboring an ABL1 mutation *in vivo*, we established murine NSCLC xenograft models using the H1915 cell line, carrying R351W homozygous mutation in ABL1, and the H650 cell line, expressing wild-type ABL1 as a control model. All mice formed engrafted tumors within 10 days and were treated orally, twice daily with imatinib (100 mg/kg) or vehicle. The mice tolerated the therapy well without any changes in body weights or signs of toxicity during the course of the treatment (Appendix Fig S4A–C). For H1915-derived xeno-grafts, we observed significant reduction in tumor volumes in the imatinib-treated group, compared to the vehicle-treated animals, from day six of the drug administration (Fig 3A). In contrast, imatinib had no effect on tumor growth in H650 xenografts (Fig 3B). Finally, to confirm the specificity of the pharmacological responses *in vivo*, imatinib was administered to mice carrying tumors derived from the H1915 cell line with doxycycline-inducible expression of drug-resistant ABL1-T315I. Overexpression of the drug-resistant ABL1 rescued the imatinib-mediated growth inhibitory effects in the H1915 xenografted tumors (Fig 3C). These data, together with the lack of response toward imatinib treatment observed in the H650 xenografts, indicate that the therapeutic effects of imatinib *in vivo* are due to specific inhibition of mutant ABL1-R351W kinase.

### ABL1 cancer mutants display differential regulation of cell cycle vs. cell survival

To determine the mechanism by which imatinib-induced ABL1 inhi-bition suppressed cell viability, we monitored changes in cell cycle regulators as well as proteins associated with apoptosis in the H1915, H2110, and H650 cells treated with imatinib. No changes were observed in the H650 cell line containing WT ABL1, but harboring the ABL2-W469C mutation (Fig EV4A). Interestingly in the H1915 cells treated with imatinib, we observed a significant increase in p21 protein levels (Fig EV4A) and this correlated with an increase in the percentage of cells in G1 phase and a decrease in the percentage of cells in S-phase of the cell cycle (Fig EV4B). For the H2110 cells treated with imatinib, we observed a decrease in p21 expression levels and an increase in cleaved caspase 3, 7, and 9 as well as an increase in cleaved PARP (Fig EV4A). This correlated with an increase in early and late apoptotic cells (Fig EV4C) indicat-ing that the G340L mutant is promoting cell survival compared to the R351W mutant that is promoting cell cycle progression. These observations are consistent with different mutant forms of oncoge-nes regulating unique downstream phenotypes (Ihle *et al*, 2012).

### Structural analysis of ABL1 mutations

To gain insight into the structural consequences of these mutations in the ABL1 kinase, molecular dynamics simulations were carried out on ABL1-WT, ABL1-G340L, and ABL1-R351W. Simulations were

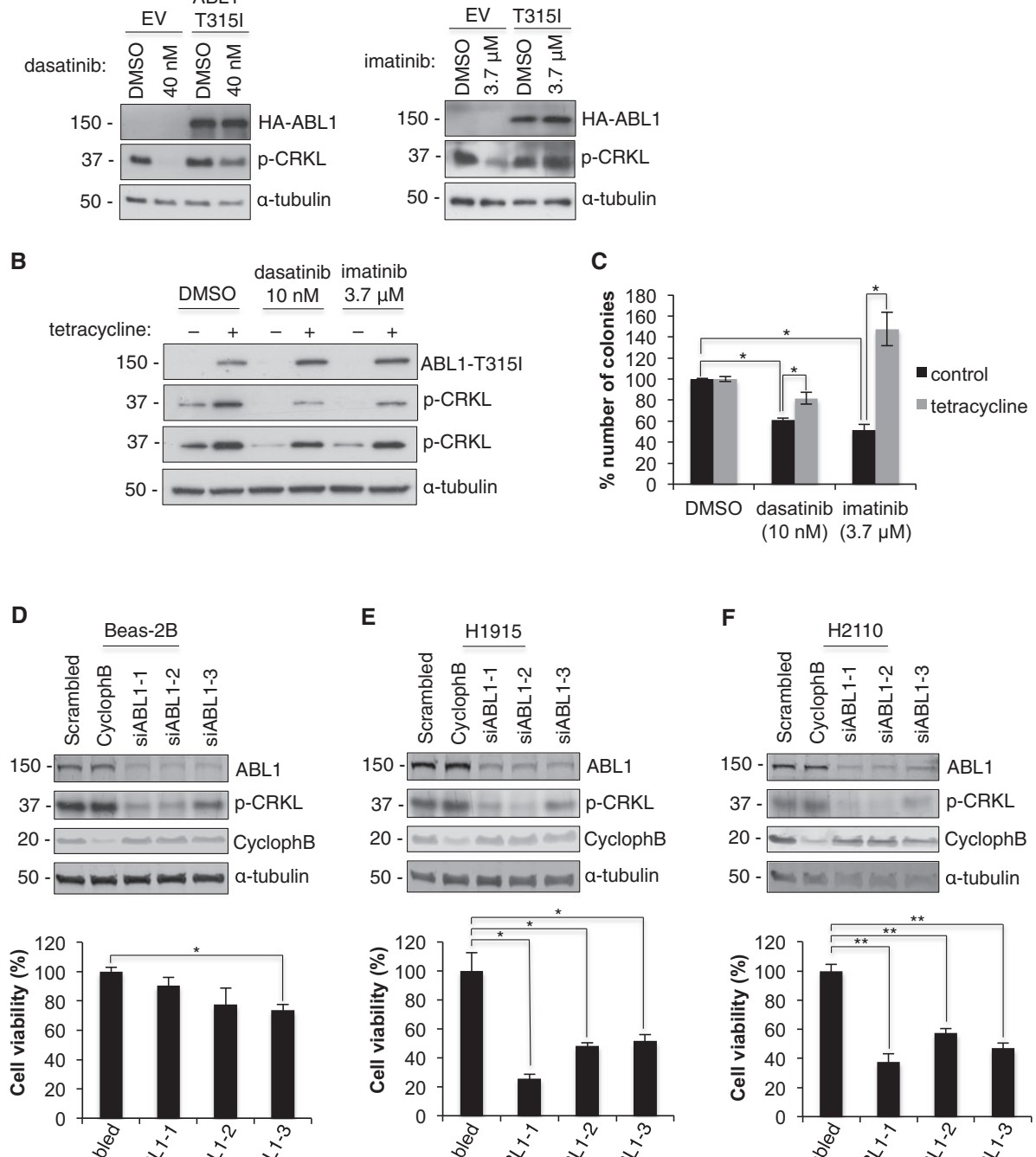

**Figure 2.  Drug-mediated effects on cell viabilities are specific to ABL1 inhibition.**

A–C  Drug-resistant ABL1-T315I rescues dasatinib- and imatinib-induced cytotoxicity. Western blot analysis of (A) transient and (B) tetracycline-inducible overexpression of ABL1-T315I in H1915 cells treated with dasatinib or imatinib. (C) CFA with dasatinib or imatinib for H1915 cells with or without tetracycline-inducible expression of ABL1-T315I. Error bars represent mean ± S.E.M., $n = 9$, *$P < 0.001$ by two-tailed unpaired Student's $t$-test (control, DMSO vs. dasatinib $P = 4.3 \times 10^{-12}$ vs. imatinib $P = 6 \times 10^{-6}$; dasatinib, control vs. tetracycline $P = 9 \times 10^{-7}$; imatinib, control vs. tetracycline $P = 9.7 \times 10^{-5}$).

D–F  Knockdown of mutated ABL1 reduces viability of lung cancer cells. Top, Western blot analysis of ABL1 knockdown in (D) Beas-2B, (E) H1915, and (F) H2110 cells upon transfection with three ABL1 siRNAs, negative control (scrambled) and (Cyclophilin B) positive control. Bottom, MTT assay showing viability of (D) Beas-2B, (E) H1915, and (F) H2110 cells in response to ABL1 knockdown. Error bars represent mean ± S.E.M, $n = 9$, *$P < 0.005$, **$P < 0.001$ by two-tailed unpaired Student's $t$-test (Beas-2B, scrambled vs. siABL1-3 $P = 0.002$; H1915, scrambled vs. siABL1-1 $P = 0.004$, vs. siABL1-2 $P = 0.001$, vs. siABL1-3 $P = 0.002$; H2110 scrambled vs. siABL1-1 $P = 5 \times 10^{-6}$, vs. siABL1-2 $P = 2 \times 10^{-5}$, vs. siABL1-3 $P = 4.2 \times 10^{-6}$).

Source data are available online for this figure.

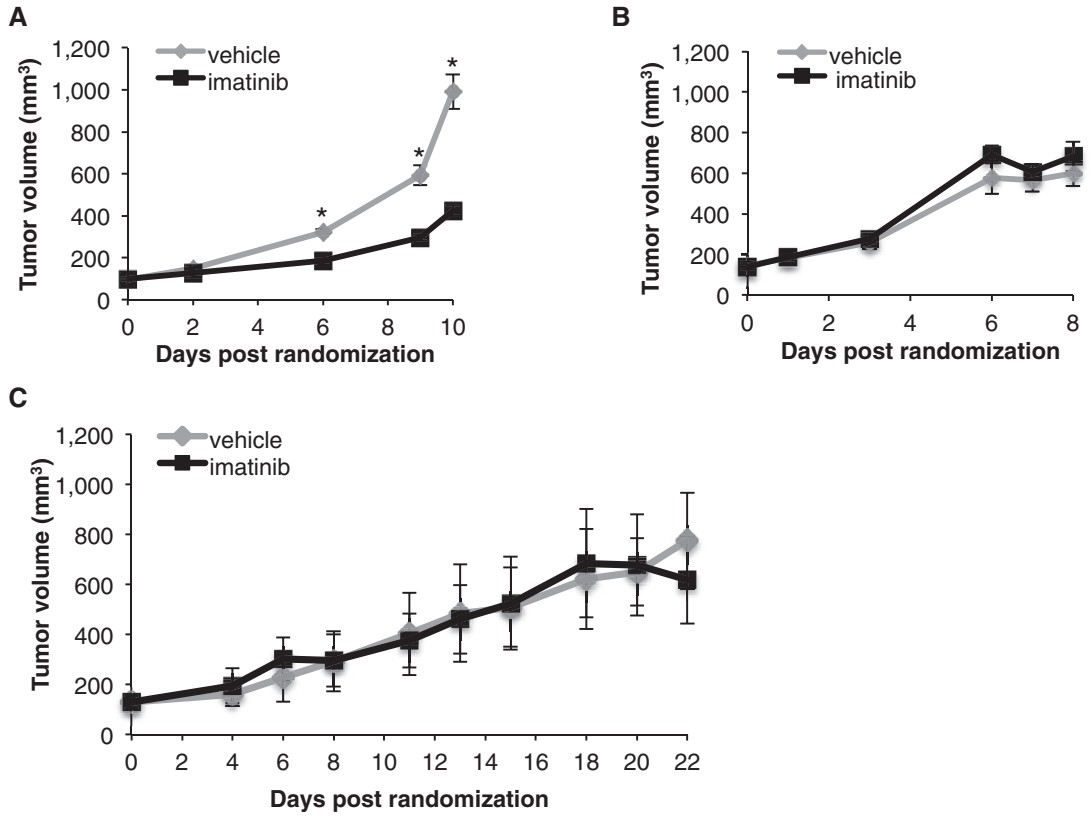

**Figure 3.  Imatinib reduces tumor growth *in vivo*.**

A–C    Average tumor volumes of xenografts derived from H1915 (A), H650 (B), and (C) H1915 cell line with doxycycline-inducible ABL1 gatekeeper mutation (H1915-ABL1$^{T315I}$) in mice treated with either vehicle or imatinib (100 mg/kg). Error bars represent ± S.D., *n* = 6–10 mice per group, *P = 0.0016 by Mann–Whitney *U*-test.

performed on active and inactive structures of the kinase domain alone, in addition to the SH3-SH2-kinase domain structure; the domain composition and structure of the ABL1 kinase are provided as a reference (Appendix Fig S5A and B). Analysis of the level of movement (root-mean-squared deviation; RMSD) detected within the constructs over the course of the simulations highlighted key changes within the αC-helix and DFG motif within ABL1-R351W (Fig 4). Specifically, the αC-helix, a region known to alter position in order to reach the active kinase conformation, was observed to have greatly increased movement within the inactive conformation of the R351W kinase domain, while no change was observed in the active conformation (Fig 4A). Time-lapse images from the simulations with the inactive construct depict the αC-helix of the R351W (orange outline) shifting in a manner consistent with the kinase progressing from an inactive conformation to an active conformation, while the αC-helix of the WT (black outline) construct remains still (Fig 4B). Consistent with these data, we also observed greater movement in the inactive ABL1-R351W DFG motif, which shifts during kinase activation to achieve correct positioning for binding to ATP. Again, no difference was detected between ABL1-R351W and ABL1-WT in the active conformation (Fig 4C). Time-lapse images from the inactive kinase conformations suggest the DFG motif is shifting toward an inward active position when the R351W mutation is present (Fig 4D). The increased movement of both of these features within the R351W inactive conformation suggests an

increased propensity for transitioning to the active kinase conformation. Furthermore, major structural differences were observed between ABL1-R351W and the WT kinase, with wide-scale destabilization of the domain structure, discussed further in the Appendix (Appendix Fig S6A–D).

Molecular dynamics simulations of the inactive kinase domains of ABL1-WT and ABL1-G340L showed a small increase in movement within the catalytic spine (C-spine) of the mutated protein. During kinase activation, the C-spine residues lock together in a solid spine spanning the two lobes of the kinase, combined with the adenine ring of the ATP molecule, creating a stable active kinase conformation (Kornev *et al*, 2008). We observed a stabilization of the C-spine within the active G340L kinase domain, compared to the wild-type construct, indicating the kinase may maintain greater stability in the active conformation (Appendix Fig S7A and B).

**ABL1 mutations in lung cancer have increased activity toward CRKL**

To assess whether mutations in ABL1 altered the activity of the kinase, R351W and G340L were overexpressed in H1915 cells. In these cells, the ABL1-R351W showed a modest increase in activity toward CRKL when compared to WT ABL1 (Appendix Fig S8). No increase in CRKL phosphorylation was observed for ABL1-G340L (Appendix Fig S8). Consistent with these results, overexpression of

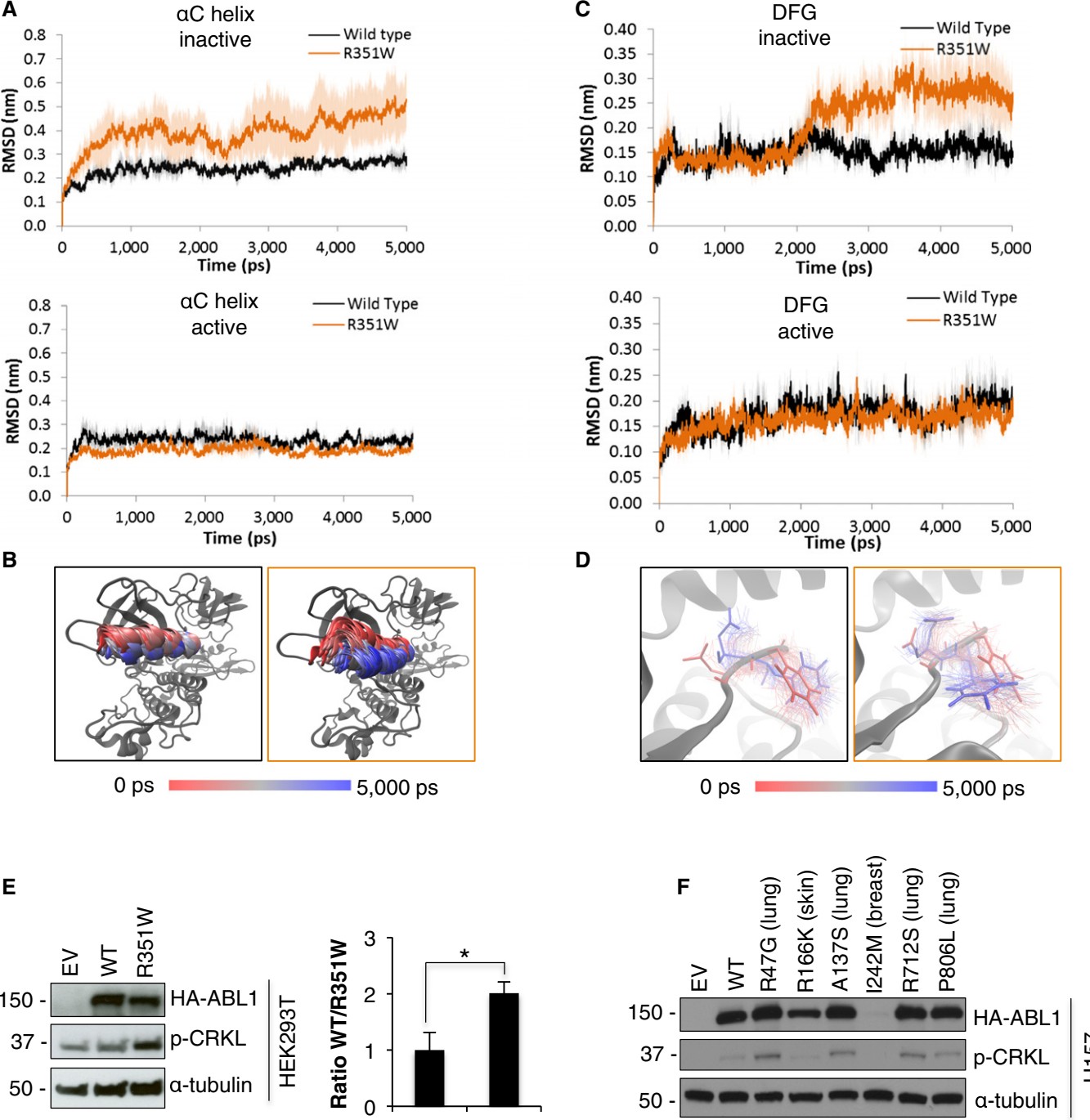

**Figure 4. ABL1 mutations lead to intermediate increase in kinase activity.**

A–D  (A, C) Root-mean-square deviation (RMSD) analysis of the movement of (A) αC-helix and (C) DFG motif in the active and inactive conformation of ABL1 kinase with R351W mutation (orange) compared to the wild-type (WT) kinase (black). (B, D) Time-lapse images show movement within the αC-helix (B) and DFG motif (D) of the inactive structure of the WT (black outline) and R351W (orange outline) constructs.

E  Left, Western blot analysis of transient overexpression of wild-type ABL1 (WT) and ABL1-R351W, compared to empty vector control (EV) in HEK293T cells. Right, quantification of p-CRKL signal based on densitometry analysis of blots from 6 independent experiments. Error bars represent ± S.D., $n = 6$, *$P = 0.0004$ by two-tailed unpaired Student's *t*-test.

F  Overexpression of mutant ABL1 constructs with mutations identified in selected primary lung tumors (Ding *et al*, 2008) in H157 cells. Values indicate quantified band intensity for p-CRKL, relative to α-tubulin and HA-tagged ABL1.

Source data are available online for this figure.

ABL1-R351W in HEK293T cells resulted in a 2-fold increase in CRKL phosphorylation when compared to ABL1-WT (Fig 4E). ABL1 mutations identified in primary tumors from the lung, breast, and skin, which were predicted to be damaging (Appendix Table S2), were assessed and the four mutated ABL1 variants identified in primary lung tumors displayed increased activity based on increased CRKL phosphorylation compared to ABL1-WT (Figs 4F and EV5).

### ABL1 mutations increase cytoplasmic localization

To assess whether the mutations alter the localization of ABL1, we compared immunofluorescence images of the endogenous ABL1 in Beas-2B cells against a panel of NSCLC cell lines. In Beas-2B, H650, and A549 cell lines, ABL1 was equally distributed between the nuclear and the cytosolic compartments. However for H1915 and H2110 cell lines, we observed significantly lower nuclear localization of ABL1, indicating that ABL1 mutations are associated with cytoplasmic retention of ABL1 (Fig 5A and B). The specificity of ABL1 staining was evaluated by immunofluorescence imaging in cells depleted of ABL1 (Appendix Fig S9).

To validate these data, we induced expression of ABL1-WT, ABL1-G340L, and ABL1-R351W constructs in the Beas-2B cells and evaluated localization by immunofluorescence. Upon induced expression, both ABL1-R351W and ABL1-G340L showed a significantly lower nuclear to cytoplasmic ratio compared to ABL1-WT (Fig 5C, Appendix Figs S10 and S11). The increased cytoplasmic retention of both ABL1 with R351W and G340L mutations is consistent with the previous reports of oncogenic ABL1 variants being predominately localized to the cytosol (Taagepera *et al*, 1998; Vigneri & Wang, 2001; Furstoss *et al*, 2002).

## Discussion

Our study highlights the need to perform comprehensive pharmacological, biochemical, and structural studies to validate potential cancer genetic dependencies. As illustrated by our bioinformatics analysis, mutations in both ABL1 and ABL2 were predicted as pathogenic. However, our data indicate that only mutations in ABL1 would be pathogenic by promoting lung cancer cell survival and proliferation. We demonstrate that lung cancer cells harboring ABL1 mutations are sensitive to clinically relevant concentrations of imatinib and dasatinib and that these effects are specific to pharmacological inhibition of the mutated ABL1 as shown by the rescue experiments with a drug-resistant variant of ABL1. Moreover, reduction in cell viability upon siRNA-mediated ABL1 depletion further highlights the importance of the mutated ABL1 in promoting survival in these NSCLC cell lines. Finally, imatinib induced reduction in tumor growth in xenografts with mutated ABL1 and simultaneous lack of response observed for xenografts with wild-type ABL1 highlights the therapeutic potential of imatinib in NSCLC patients with ABL1 mutations. However, responses were not complete and likely patients with ABL1 mutations would need to be treated with a combination of imatinib and other forms of chemotherapy, which will be investigated in future studies. These results also highlight that somatically mutated ABL1 will likely make a better biomarker to predict response compared to loss of ABL1-negative regulators, such as FUS1, based on the lack of response of the A549 cells to imatinib and ABL1 depletion.

The structural modeling and biochemical analysis indicated that mutations in ABL1 lead to an intermediate increase in kinase activity toward CRKL. Gain-of-function mutations often enhance or confer tumorigenic phenotypes and somatically mutated ABL1 is likely contributing to enhanced survival and/or proliferation of cancer cells through increased activity. Interestingly, the ABL1-G340L did not display increased activity toward CRKL and this may be due to the mutation promoting stabilization of the active form of the kinase, as indicated by the structural modeling of the ABL1-G340L (Appendix Fig S7), rather than shifting the kinase into an active conformation, as observed for the R351W. However, we did observe an increased cytosolic localization of both ABL1-G340L and ABL1-R351W in the H2110 and H1915 cells and when overexpressed in Beas-2B cells. ABL1-WT is known to shuttle between the nucleus and cytosol where it performs distinct functions; nuclear localization is known to be associated with pro-apoptotic effects, while cytosolic localization has been implicated in mitogenic properties (Taagepera *et al*, 1998; Vigneri & Wang, 2001). In addition, oncogenic forms of ABL1 (including BCR-ABL) are retained in the cytosol, where they can activate cell growth and anti-apoptotic pathways (Taagepera *et al*, 1998; Vigneri & Wang, 2001). Therefore, increased cytoplasmic localization of ABL1-R351W and ABL1-G340L could have a similar function and contribute to the increased survival of lung cancer cells harboring these mutations (Furstoss *et al*, 2002). Interestingly, we observed the G340L promoted cell survival in H2110 cells, likely due to an altered localization and regulation of pro-survival pathways, whereas the activated R351W promoted cell cycle progression, which could be attributed to constitutive activation of this mutant. The study of additional ABL1 mutants will shed light on this possibility and discern if this is more generally applicable.

There is an urgent need for the identification of actionable somatic mutations in patients with lung cancer (Kim *et al*, 2011). As clinical trials become ever more personalized, genetic biomarkers are being utilized for patient stratification and treatment decisions (Kim *et al*, 2011). Here, we provide data that supports a role for mutationally altered ABL1 as a novel genetic dependency in a subset of patients with lung cancer. Consistent with our data, CRKL was recently identified as a novel lung cancer driver, found on a common focal amplification (Cheung *et al*, 2011). This further implicates increased signaling in the ABL-CRKL signaling pathway as an important survival event for lung cancer cells. Lastly, cancer genomic studies indicate that 1–2% of patients with lung adenocarcinoma will have mutations in ABL1. Therefore, imatinib or dasatinib could represent a treatment option for 8,500–17,000 patients with lung cancer globally that harbor mutations in ABL1 (http://globocan.iarc.fr/old/FactSheets/cancers/lung-new.asp). Overall, this work demonstrates that somatically mutated ABL1 represents a genetic dependency that could be exploited therapeutically for a subset of patients with lung cancer.

## Materials and Methods

### Cell culture

H2110, H1915, H650, H1623, H2126, HCC827, A549, H810, H1048, H1355 and HEK293T, and Beas-2B were purchased from ATCC. The H157 cell line was a kind gift from Dr Phillip Dennis (Johns Hopkins

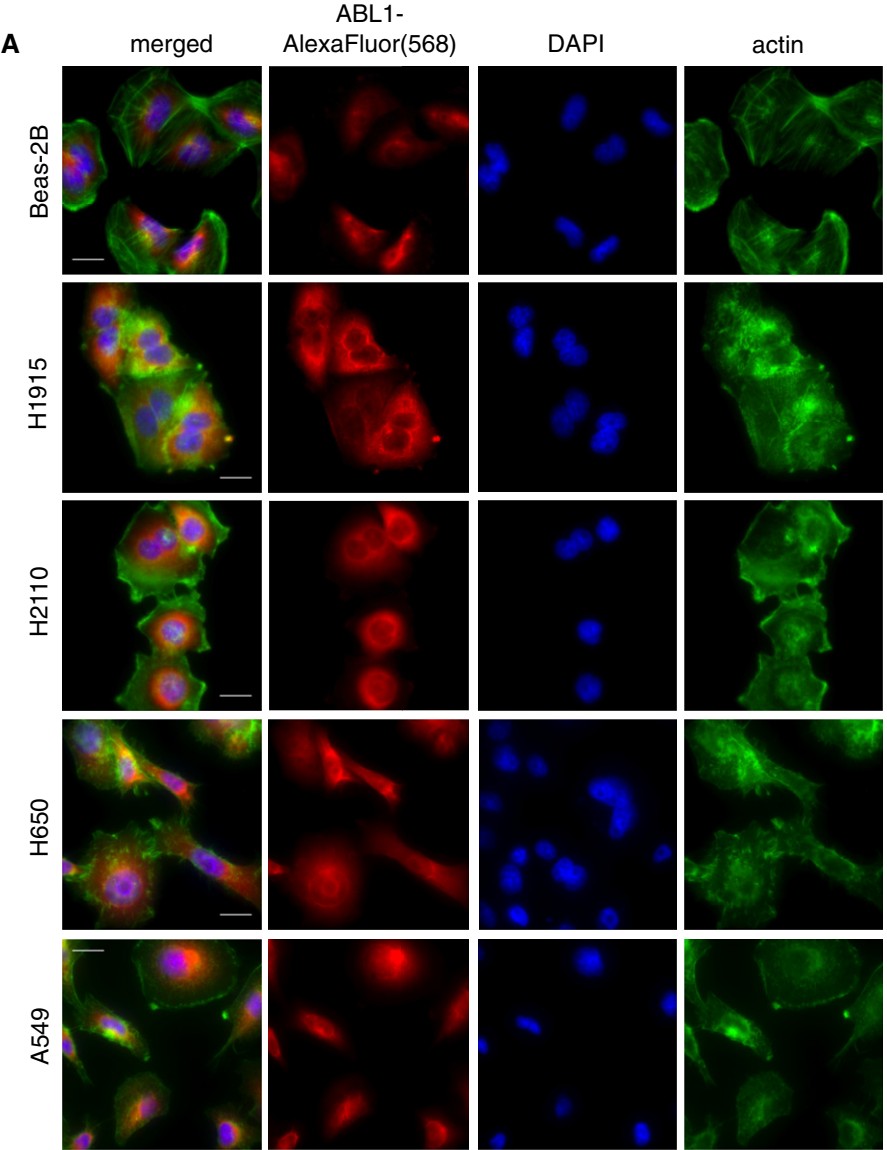

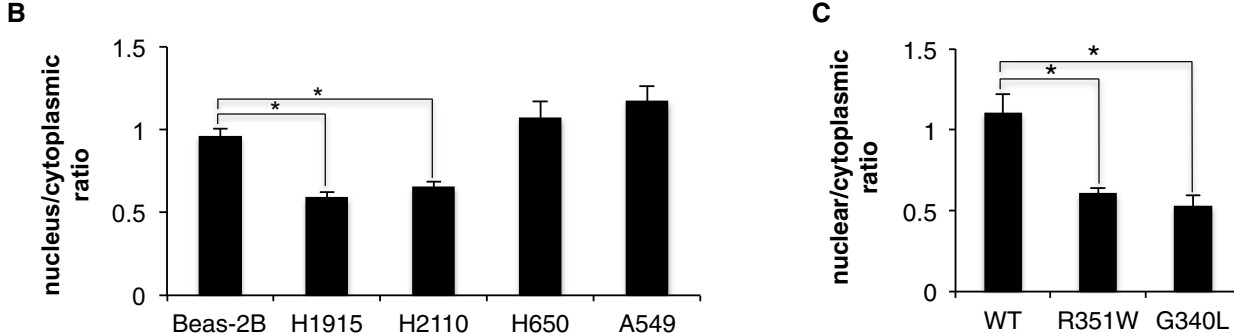

**Figure 5.  ABL1 mutations increase cytosolic retention of the mutated ABL1 proteins.**

A     Immunofluorescence imaging of endogenous ABL1 in the panel of NSCLC cell lines; scale bars = 20 μm.

B, C   Quantification of nuclear vs. cytoplasmic localization of (B) endogenous ABL1 in NSCLC cell lines and (C) tetracycline-inducible expression of ABL1-WT, ABL1-R351W, and ABL1-G340L in Beas-2B cells. Error bars represent ± S.D., n = 6–10 images, *P < 0.005 by two-tailed unpaired Student's t-test (Beas-2B vs. H1915 $P = 6.6 \times 10^{-5}$, vs. H2110 P = 0.0001; WT vs. R351W P = 0.003, vs. G340L P = 0.002).

University). All cell lines were maintained in RPMI1640 media (Invitrogen) supplemented with 2 mM L-glutamine (Gibco), 10% FBS (Lonza), and 1% Pen/Strep (Gibco). 293FT cells (Invitrogen) were maintained in DMEM media (Gibco) supplemented with 10% FBS (Lonza) 4 mM L-glutamine (Gibco), 1 nM sodium pyruvate (Gibco), and 0.1 mM NEAA (Gibco). All experiments were performed within 6 months of cell line delivery and cell lines were authenticated by DNA sequencing at the start of experiment. After 6 months, cells were discarded and new cells ordered from ATCC. Mycoplasma testing was performed on regular basis.

## Inhibitors and MTT assay

Imatinib (STI571, Gleevec) and dasatinib (BMS-354825) were purchased from Selleck Chemicals. The compound doses chosen for the current study were based on 1:3 serial dilution dose curve starting at 12 μM. For Western blotting, the compounds were administered to cells 1 h prior to lysis at indicated doses. The 72-h MTT assays were performed in 96-well format with imatinib or dasatinib administered in triplicate at indicated inhibitor concentrations. Cell viabilities were assayed using a MTT Kit (3-(4,5-Dimethylthiazole-2-yl)-2,5-diphenyltetrazolium bromide) (Sigma), according to the manufacturer's instructions. The plates were read on a Bio-Rad iMark Microplate Reader at 595 nm. At least three independent MTT experiments were carried out ($n = 9$). The data were normalized to control DMSO-treated cells.

## siRNA-mediated silencing

The cells were reverse-transfected with 50 nM of appropriate siRNA, using Lipofectamine 2000 (Invitrogen), according to the manufacturer's instructions. The following siRNAs were utilized in the study: siABL1-1 (RI10035, Abgent), siABL1-2 (SASI_Hs01_00174552, Sigma), siABL1-3 (SASI_Hs01_00154491, Sigma), siABL2-1 (1478, Life Technologies), siABL2-2, (1662, Life Technologies), siABL2-3 (SMARTpool On-Target plus ABL2 siRNA, Thermo Scientific), On-Target plus cyclophilin B (D-001820-01-20, Thermo Scientific), and On-Target plus Non-targeting siRNA #1 (D-0018110-01-20, Thermo Scientific). The protein knockdown was assessed at 72 h post-transfection by Western blotting, while the cell viability was assessed by MTT assay, as described above, at 4 days post-transfection. Data were normalized to non-targeting siRNA.

## Transient overexpression

HEK293T and H157 cells were transfected with the ABL1 constructs and the empty vector control using Attractene transfection reagent (Qiagen), according to the manufacturer's instructions. For overexpression in H1915 cells, the cells were subjected to reverse transfection with the ABL1 constructs, using Lipofectamine 2000 (Invitrogen), according to the manufacturer's instructions. Protein overexpression was assessed at 48 h post-transfection by Western blotting.

## Generation of tetracycline-inducible cell lines

ABL1-WT, ABL1-R351W, ABL1-G340L, ABL1-R47G, ABL1-A137S, ABL1-R712S, ABL1-P806L, and gatekeeper ABL1-T315I genes were cloned into pLenti/TO/V5-DEST vector from the respective

pDONR223-ABL1 constructs, using LR clonase reaction, as described above. 293FT cells were seeded on 6-well plates at approximately 80% confluency. Following 24-h incubation, the cells were transfected with the ABL1 viral constructs or pLenti3.3/TR vector (for tetracycline repressor (TR) expression) using the Vira-Power HiPerform T-Rex Gateway Expression System (Invitrogen) and Lipofectamine 2000 (Invitrogen), following the manufacturer's instructions. The lentiviral stocks were collected 4 days later. H1915 and Beas-2B cells were then transduced first with TR lentiviral stock and then with the ABL1 lentiviral stocks on two consecutive days in the presence of 6 μg/ml polybrene (Sigma). Cells expressing both constructs were selected for over 10 days using 10 μg/ml blasticidin (Invitrogen) and 500 μg/ml geneticin (Gibco). A total of 1 μg/ml of tetracycline (Invitrogen) was used to induce expression of wild-type ABL1 and the mutant variants of ABL1 constructs in either Beas-2B or H1915 cells.

## Colony formation assay

The cells were seeded on 6-well plates at a density of 500–1,000 cells/well, according to the cell line. The cells were dosed with the relevant inhibitor or DMSO in triplicate and incubated for 14 days. For the inducible Beas-2B cell lines, the tetracycline was added every 48 h at 1 μg/ml. Colonies formed were then washed with PBS and fixed with ice-cold methanol for 10 min. Fixed colonies were stained with 0.5% crystal violet, washed with water and air-dried. Crystal violet was dissolved in 10% acetic acid with shaking for 1 h and the absorbance measured at 595 nm. At least three independent CFA experiments were carried out ($n = 9$). The data were normalized to control DMSO-treated cells.

## Western blotting

Cells were lysed with RIPA buffer (Sigma, R0278) supplemented with protease inhibitor cocktail (Roche, 05 056 489 001) and phosphatase inhibitors (Sigma, P2850 and P5726). Lysates were resolved by SDS–PAGE followed by Western blotting. Primary antibodies: ABL1 (K-12, sc131, Santa Cruz); phospho-CRKL (Tyr207, 3181); from Cell Signaling: phospho-SRC (Tyr416, 2101), p21 (2946), PARP (9532), cleaved caspases 3 (9661), 7 (8438) and 9 (9501); phospho-FAK (Tyr576, Invitrogen), HA (MMS101R, Covance); from Abcam: cyclophilin B (Ab16045) and ABL2 (Ab54696); α-tubulin (T9026, Sigma). Secondary antibodies: HRP-conjugated mouse or rabbit (Cell Signaling). α-tubulin was used as a loading control for all Western blot experiments, and all blots displayed are representative of three independent experiments. The intensity of the bands was calculated using ImageJ, and the data were normalized to α-tubulin and to HA-tagged ABL1 constructs and presented as ratio of ABL1-WT vs. mutated variants of ABL1.

## Immunofluorescence

The cells were seeded at 70–80% confluency onto glass coverslips for 24 h. For ABL1 knockdown, the cells were subjected to reverse transfection with siABL1-2 siRNA, as described above. The inducible Beas-2B cell lines were induced with tetracycline for 24 h. The cells were fixed with 4% formaldehyde for 10 min and permeabilized with

0.5% Triton in PBS for 15 min. Following 1-h blocking with 1% BSA, the cells were probed with 1:100 ABL1 antibody (K-12, Santa Cruz) for 1 h, washed with PBS, and probed with 1:500 Alexa Flour(568)-Anti-Rabbit secondary antibody and Alexa Fluor (488) Phalloidin (Life Technologies) for 1 h. The coverslips were mounted using Prolong Gold antifade reagent with DAPI (Life Technologies). The cells were visualized using Zeiss Low Light Microscope with 100×/1.45/oil objective. At least 6 images were taken from each independent experiment. The nuclear and cytoplasmic ABL1 localization was quantified using ImageJ, and the average values were obtained taking into account multiple images from each experiment.

**Mouse xenografts and *in vivo* drug studies**

All procedures involving animals were approved by CRUK Manchester Institute's Animal Welfare and Ethical Review body in accordance with the Animals Scientific Procedures Act 1986 and according to the ARRIVE guidelines (Kilkenny *et al*, 2011) and the Committee of the National Cancer Research Institute guidelines (Workman *et al*, 2010). Animals were group-housed in individually ventilated cages, with 12-h:12-h light/dark cycle and fed with TekLad 2019 diet. H1915, H650, and H1915-ABL1$^{T315I}$ cells were resuspended in 1:1 mixture of serum-free media and growth factor reduced matrigel matrix (BD Biosciences). Four- to six-week-old female SCID-*bg* mice (Harlan) were injected subcutaneously into the right flank with 200 μl containing $5 \times 10^6$ of cells ($n = 30$ mice per each cell line). Tumor formation was monitored every two days and tumor volume based on caliper measurements was calculated by the formula: tumor volume = (length × width × width)/2. When the tumors reached 150–250 mm$^3$, the mice were randomly allocated to groups of ten animals, without blinding. For the H1915-ABL1$^{T315I}$-derived xenografts, the mice were placed on the doxycycline diet (Mod LabDiet® 5002 with 625 ppm Doxycycline) immediately after the randomization and maintained on the doxycycline diet until the end of the experiment. Imatinib stock solutions of 10 mg/ml were prepared daily in sterile-filtered water and administered by gavage twice daily every 6–8 h at a dose of 100 mg/kg. The control mice were treated with the same volume of vehicle (sterile water). Mice were culled when tumors reached max-permitted volume of 800–1,200 mm$^3$. Animals were excluded if deterioration in clinical condition was observed or body weight dropped below 15%.

**Cell cycle and apoptosis analysis by FACS**

For the cell cycle analysis, the cells were stained with propidium iodide using Cycle TEST PLUS DNA Reagent Kit (Becton Dickinson). For apoptosis analysis, the cells were stained for annexin V-APC and 7-AAD following the manufacture's protocol (BD biosciences). The cells were analyzed by flow cytometry using FACSCalibur (Becton Dickinson). The data were analyzed using FlowJo version 10.0.8.

**Statistical analysis**

Statistical analysis was done with GraphPad Prism 5 software. For MTT, CFA, and quantification of quantified immunofluorescence imaging, statistical comparison between data sets was performed using two-tailed unpaired Student's *t*-test. Data are presented as mean ± S.E.M. or ± S.D. of three independent experiments. The

---

**The paper explained**

**Problem**

Non-small cell lung cancer (NSCLC) accounts for 80% of lung cancers affecting 1.37 million patients each year worldwide. The molecular mechanisms underlying approximately 50% of NSCLCs remain unknown, and only a small proportion of patients with lung cancer benefit from targeted therapies. Hence, there is a need to identify novel driver mutations that can be targetable in patients with lung cancer.

**Results**

Here, we identify somatically mutated ABL1 as a novel driver in lung cancer, where cells with *ABL1* mutations are sensitive to ABL1 inhibitors, imatinib and dasatinib. Expression of a drug-resistant mutant of ABL1 fully rescues the drug-induced cell death and siRNA-mediated depletion of ABL1 significantly reduces cell viability, validating the specificity of ABL inhibitors and highlighting the importance of mutationally altered ABL1 in lung cancer cells. Structural modeling indicated that the ABL1 mutations would shift critical regions of the kinase into a more active conformation. Consistent with our modeling, we demonstrate that mutations in ABL1 in primary lung tumors and a lung cancer cell line increase the activity of ABL1 and mutated ABL1 proteins are primarily localized to the cytosol. Lastly, we show antitumor activity of imatinib in xenografts derived from NSCLC cell line harboring an ABL1 mutation.

**Impact**

Since approximately 1–2% of patients with lung adenocarcinoma harbor ABL1 mutations, dasatinib or imatinib could represent a treatment option for 8,500–17,000 patients with lung cancer globally.

---

variance was determined by *F*-test. For xenograft studies, the data did not fit normality as tested by Kolmogorov–Smirnov test and statistical significance was obtained by Mann–Whitney *U*-test and data are presented as ± S.D.

Cell line sequencing and site-directed mutagenesis and cloning are described in the Appendix Materials and Methods.

**Expanded View** for this article is available online.

### Acknowledgements

This research was fully supported by Cancer Research UK. We thank members of the Signalling Networks in Cancer Group for critical review of the manuscript.

### Author contributions

ET and JB conceived the project and designed all research experiments; ET, PT-A, AAM, EWT, and AH performed experiments; NLS, ET, and JB performed structural analysis and/or interpreted results, ET, NLS, PT-A, AAM, and JB wrote and prepared the manuscript and supplemental information. CLH, CJM, and CD assisted in the *in vivo* study.

### Conflict of interest

The authors declare that they have no conflict of interest.

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
