## [Review Process File · EMBO Molecular Medicine]

Somatically Mutated ABL1 is an Actionable and Essential NSCLC Survival Gene

Ewelina Testoni, Natalie L. Stephenson, Pedro Torres-Ayuso, Anna A. Marusiak, Eleanor W. Trotter, Andrew Hudson, Cassandra L. Hodgkinson, Christopher J. Morrow, Caroline Dive and John Brognard

Corresponding author: John Brognard, Cancer Research UK Manchester Institute

Review timeline:

Submission date:	09 September 2014
Editorial Decision:	15 September 2014
Appeal:	16 September 2014
Editorial Decision:	18 September 2014
Resubmission:	21 May 2015
Editorial Decision:	25 June 2015
Revision received:	04 November 2015
Editorial Decision:	17 November 2015
Revision received:	25 November 2015
Accepted:	08 December 2015

Editor: Roberto Buccione

Transaction Report:

1st Editorial Decision

15 September 2014

Thank you for the submission of your manuscript "Somatically Mutated ABL1 Represents an Actionable and Essential Lung Cancer Survival Gene".

I have now had the opportunity to carefully read your paper and the related literature and I have also discussed it with my colleagues. I am afraid that we agreed that the manuscript is not well suited for publication in EMBO Molecular Medicine and have therefore decided not to proceed with peer review.

You find that dasatinib and imatinib reduced CRKL phosphorylation and viability in lung cancer cell lines with ABL1 mutations and that expression of a drug-resistant ABL1 mutant rescued, whereas ABL1 knock-down reduced viability. We appreciate that you carry out an extensive structural analysis of ABL1 mutants and that the mutants do have increased phosphorylation activity towards CRKL.

Although we acknowledge the potential interest of your findings, we are not persuaded, due to the in vitro nature of your approach, that your manuscript provides the level of translational development we would like to see in an EMBO Molecular Medicine article.

Please rest assured that this is not a judgement of the quality and value of your work but a decision based on suitability for EMBO Molecular Medicine in its current form. In fact, we would be pleased to consider a future version that would provide an in vivo proof of concept, for example with a PDX mouse model with ABL1 mutations.

I am sorry to have to disappoint you at this stage.

Appeal

16 September 2014

Thank you very much for your response and considering our manuscript. I wanted to inquire and determine if utilising a cell line based (H1915) xenograft mouse model where we can induce expression of drug resistant ABL1, would be acceptable for in vivo data? This would demonstrate that in the in vivo setting the mutant ABL1-R351W is specifically required to maintain lung cancer cell survival and the effects are specific to ABL1 inhibition as the drug-resistant mutant may be able to overcome imatinib or dasatinib induced tumour regression. For the PDX model, the frequency of the ABL1 mutations is lower and would require acquisition and sequencing of approximately 100 NSCLC patient samples and then hoping the 1 or 2 that would have ABL1 mutations would take in a mouse and this would likely take 1-2 years to generate this data. Thank you for reviewing and considering my inquiry. I look forward to your response.

2nd Editorial Decision

18 September 2014

Thank you for your letter regarding our recent decision on your manuscript entitled "SOMATICALLY MUTATED ABL1 REPRESENTS AN ACTIONABLE AND ESSENTIAL LUNG CANCER SURVIVAL GENE" and your continued interest in publishing your work EMBO Molecular Medicine.

I perfectly understand the point you make that to create from scratch an ABL1 mutation-carrying PDX model (and then collect the data of course) would not be a short-term venture, and indeed my suggestion was rather based on the assumption that you might have been already setting up such a model.

That said, and after discussion, we would be willing to reconsider a revised manuscript incorporating additional experimentation along the lines you suggest. Specifically, if you could validate, in a xenograft (ideally orthotopic) mouse model, your findings concerning both the sensitivity of the ABL1 mutants to imatinib and the refractory behavior of the drug-resistant ABL1 mutant, I would commit to sending out your manuscript for peer-review.

Please let me know by return email if you wish to move forward with your manuscript as suggested above.

I look forward to seeing a revised form of your manuscript as soon as possible.

Resubmission

21 May 2015

Please find enclosed an updated version of our manuscript entitled "SOMATICALLY MUTATED ABL1 REPRESENTS AN ACTIONABLE AND ESSENTIAL NSCLC SURVIVAL GENE" where we have now included a xenograft mouse model that demonstrates sensitivity to imatinib in Figure 3H. We observe that mice xenografted with the H1915 lung cancer cell line harbouring an R351W GOF mutation in ABL1 are sensitive to treatment with imatinib compared to a control cell line H650 (figure 3H). We now hope that our manuscript will be deemed suitable for review at EMBO Molecular Medicine and it prepared as a Report.

Thank you for the submission of your manuscript to EMBO Molecular Medicine. We are sorry that it has taken longer than usual to get back to you on your manuscript. In this case we experienced some difficulties in obtaining the Reviewer evaluations in a timely manner. Further to this, I wished to discuss the evaluations with my colleagues.

As you will see the three Reviewers are globally positive. However, the issues raised are few but quite important. Although I will not dwell into much detail, I would like to highlight the main points.

Reviewer 1 would like you to provide more details on treatments, including drug dose-response curves and would also like more information on the effects of the ABL1 mutants on other cellular parameters (see also comments from Reviewer 2). Finally, s/he feels that the in vivo experimentation requires validation and extension, for example by testing the gatekeeper mutant as well. Reviewer 1 also asks a number of very interesting questions concerning the dataset that should not prove to difficult to address and would add value and impact to the conclusions.

Reviewer 2 raises, in our opinion, a most crucial issue here. S/he notes that the effects obtained targeting mutant ABL1 are statistically significant but biologically modest and thus deserving more in-depth analysis to establish efficacy, one way or the other. The Reviewer 2 suggests a number of experiments aimed at establishing the potential for a larger effect of imatinib by exploiting more sensitive tests. S/he also notes that ideally, it would be of considerable interest to investigate imatinib resistance.

Reviewer 3 is also supportive and would like you to show the impact of mutations on Abl kinase activity through appropriate kinase assays.

In conclusion, while publication of the paper cannot be considered at this stage, given the potential interest of your findings and after internal discussion, we have decided to give you the opportunity to address the above concerns.

We are thus prepared to consider a substantially revised submission, with the understanding that the Reviewers' concerns must be addressed with additional experimental data where appropriate and that acceptance of the manuscript will entail a second round of review. I would suggest addressing Reviewer 1's comments in full and to also follow-up on Reviewer 2's request to extend your investigations to additional biological assays (which would also address some of Reviewer 1's requests). As for Reviewer 2's other, pertinent comments, and Reviewer 3's request for kinase activity analysis, I do encourage you to develop your study as far as realistically possible with the understanding that addressing these concerns would not be considered mandatory. We also suggest, pending the outcome of your further experimentation, that you eventually discuss the modest outcome of the preclinical findings, in the manuscript. The overall aim is to significantly upgrade the clinical relevance and usefulness of the dataset, which of course would in turn increase the broader impact of your work.

I understand that if you do not have the required data available at least in part, to address the above, this might entail a significant amount of time, additional work and experimentation and might be technically challenging, I would therefore understand if you chose to rather seek publication elsewhere at this stage. Although we hope not, should you do so, we would welcome a message to this effect. Also, please do not hesitate to contact me regarding the revision at any point in time.

***** Reviewer's comments *****

Referee #1 (Comments on Novelty/Model System):

This is a very interesting paper evaluating if ABL1 and ABL2 mutations found in NSCLC may be new driver mutations that can be targetable in lung cancer patients.

Referee #1 (Remarks):

This is a very interesting paper evaluating the potential therapeutic value of ABL1 mutations found in NSCLC cell lines and patients. My comments about the study are as follows:

Major comments:

1. Are the ABL1/ABL2 mutations found in primary tumors repetitively found in the kinase domain? A scheme of the genes with the already known mutations would be useful.
2. As in the case of CML, has any point mutation conferring resistance in ABL1 or ABL2 been described in the literature? Has ABL1 and/or ABL2 expression any correlation with prognosis in NSCLC patients? Those data could be included in a more detailed introduction. Moreover, the introduction could be more informative if a brief description of ABL pathway and function is described.
3. What is the mutational profile of the cell lines analyzed? Are other kinases such as SRC amplified in those cell lines?
4. Is there any relationship between ABL mutations and the histological subtypes of lung cancer?
5. Have authors evaluated if there is CNV of these genes? It seems that this mechanism may be important in the case of ABL2. The overexpression of ABL2 in normal cells may help to elucidate if ABL2 overexpression has a functional role in lung cancer and may serve as a potential drug target. ABL2 knockdown in tumor cells by siRNA would also help to clarify this point.
6. Authors use a treatment with 40 nM and 123 nM dasatinib. GI50 value determination and dose-response curves would be useful to determine the kinetics of the treatments. The same comments are applicable to imatinib results.
7. What is the effect of dasatinib on colony formation? Conversely, what is the effect of imatinib treatment on cell viability? had these compounds any effect on apoptosis?
8. What is the effect of the induced expression of mutated forms of ABL1 in the proliferation and cell survival of the Beas-2B cells? Did the expression of these mutated forms induce the malignant phenotype of non-tumor cells? Soft agar colony formation assays would help to elucidate the ability of cellular anchorage-independent growth.
9. What is the effect of mutations found in primary tumors on proliferation and survival of non-tumor cells?
10. Authors conclude that ABL1 mutations increase cytoplasmic localization. It would be worthy to extend this study to all the panel of cell lines included in this study. Did authors perform specific controls of the immunohistochemical technique?
11. Authors have based their "in vivo" results on one ABL1 mutated cell line. These results need to be confirmed with the inclusion of other cell lines with different mutations. The inclusion of a group with the gate-keeper construction ABL1-T315I would help to clarify the results.

Minor comments:

The percentage of NSCLC patients in the "Introduction" and "The paper explained" sections should be the same.

Referee #2 (Remarks):

EMM-2015-05456

Testoni et al. "SOMATICALLY MUTATED ABL1 REPRESENTS AN ACTIONABLE AND ESSENTIAL NSCLC SURVIVAL GENE"

The authors study the infrequently mutated genes ABL1 and ABL2 as potential therapeutic targets in non-small cell lung cancer (NSCLC). They find NSCLC cell lines with ABL1 mutations (a few lines) and those with ABL1 mutations are sensitive to the ABL kinase inhibitor imatinib, in vitro and in vivo (xenograft). Those lines with wild type ABL1 and ABL2 mutations are not sensitive. They provide evidence that NSCLCs lines with ABL1 mutations have activated known downstream targets such as pCRKL. Structural modeling studies of the mutant ABL1 protein are also provided. They conclude mutant ABL1 is a new therapeutic target for NSCLC. No clinical therapeutic trials in patients with mutated ABL1 are presented. Besides pharmacologic data on growth inhibitory effect

targeting ABL1, siRNA mediated knockdown studies of ABL1 are presented. However, no "rescue" experiments for these siRNA studies are presented to show the siRNA effect was "on target."

Comments to the authors.

This is a technically well-done study. Despite all of the structural and functional studies, the key issues for this to be clinically important are first the quantitative nature of growth inhibition of ABL1 mutant NSCLCs, and then the nature of any acquired resistance mechanisms. With regard to the quantitative effects, they appear to be quite modest compared to other mutant tyrosine kinase targeted therapy in NSCLCs such as targeting mutant EGFR or targeting BCR-ABL in CML. The hard, cold facts are in Figure 1C with an ~50-60% inhibition of colony formation, not even one log and in Figure 3H, Showing a very modest xenograft effect.

If we knew that even anecdotal patients had clinically significant and beneficial responses, these data would be fine. Alternatively, if they had tested the drug in a patient with mutant ABL1 and there were not such effects, that would be fine also, we would know the answer and see that the modest preclinical studies were actually informative about the lack of clinical effects.

If I were the authors, I would try to do some studies showing effect on things like lung cancer stem cells, test of passage as a xenograft after treatment to demonstrate that tumorigenic potential was significantly reduced, or tests of re-plating of tumor cells from colonies after imatinib treatment to show with these more sensitive tests, there was the potential for a large therapeutic effect of imatinib treatment. In addition, if they had developed an imatinib resistant variant and potentially showed the mechanism of resistance was similar or different than that found in CML it would have been interesting no matter what the result was.

Referee #3 (Comments on Novelty/Model System):

The work could benefit from some quantitative biochemistry to test the impact of the NSCLC mutations on the catalytic properties of Abl kinase activity.

Referee #3 (Remarks):

This work characterizes several novel mutations in Abl1 in NSCLC cells lines and identifies them as possible targets for therapeutic intervention in cancer. The work is for the most part rigorous and well done.

The only major limitation of this work stems from the lack of comprehensive characterization of impact of the Abl1 mutations on the biochemical activity of Abl. This analysis was limited to assessing the impact of mutant Abl1 overexpression on levels of CrkL phosphorylation and to assessing the relative partitioning of the Abl1 mutants to cytoplasm vs. nucleus. There was some speculation on the potential effects of the mutations on conformational dynamics, but this was based entirely on molecular dynamics simulation. A much more straightforward and direct method to address the impact of these mutations would be to express them in cells and measure the k_{cat} and K_m of the mutants for CrkL in vitro. This would indicate whether and how the mutations directly impact kinase activity.

1st Revision - authors' response

04 November 2015

RESPONSES TO REVIEWERS

Referee #1 (Remarks):

This is a very interesting paper evaluating the potential therapeutic value of ABL1 mutations found in NSCLC cell lines and patients.

Response: We thank the reviewer for this positive feedback and are pleased the reviewer found our manuscript interesting.

My comments about the study are as follows:

Major comments:

1. Are the ABL1/ABL2 mutations found in primary tumors repetitively found in the kinase domain? A scheme of the genes with the already known mutations would be useful.

Response: We thank the reviewer for this comment and have now added an illustration to highlight where the somatic mutations characterized in our manuscript occur in the ABL1 kinase. For the cell lines, these mutations do occur in the kinase domain, however mutations observed in primary tumors occur throughout the gene. All but one of the mutations that occur in NSCLC adenocarcinomas promote increased activation of the downstream substrate CRKL (Figs 4E-F, EV1, EV5, and Appendix Fig S8).

2. As in the case of CML, has any point mutation conferring resistance in ABL1 or ABL2 been described in the literature? Has ABL1 and/or ABL2 expression any correlation with prognosis in NSCLC patients? Those data could be included in a more detailed introduction. Moreover, the introduction could be more informative if a brief description of ABL pathway and function is described.

Response: We thank the reviewer for these suggestions. To our knowledge the gatekeeper mutation (T315I) or mutations that confer resistance have not been observed in lung cancer patients. In general the sample size is too small at this stage to assess if ABL1 or ABL2 could be used as a prognostic factor in NSCLC patients, however there is a slight trend that patients with ABL2 amplifications survive longer (Reviewer Fig 1, below).

Reviewer Figure 1

Overall Survival Kaplan-Meier Estimate SVG PDF

	#total cases	#cases deceased	median months survival
Cases with Alteration(s) in Query Gene(s)	30	9	52.56
Cases without Alteration(s) in Query Gene(s)	184	61	41.33

Reviewer figure 1 Overall survival analysis for ABL2 altered vs no alteration in TCGA lung adenocarcinoma data.

We have expanded the introduction to include a brief description of the ABL pathway and function. Highlighted below are the new additions to the introduction:

“The BCR-ABL oncoprotein is the primary driver of chronic myelogenous leukemia (CML) (Druker, 2008; Hunter, 2007). The characteristic feature of CML is the presence of the Philadelphia chromosome, the product of the reciprocal translocation between chromosomes 9 and 22 (Druker, 2008; Hunter, 2007). The product of this translocation is the BCR-ABL fusion protein, where the ABL tyrosine kinase domain is constitutively active (Hunter, 2007). Inhibition of the kinase activity by small molecule inhibitors, such as imatinib, results in robust tumor responses in CML patients carrying this translocation (Druker, 2008; Druker et al, 1996). This crucial discovery has helped usher in an era of personalized precision medicine aimed at inhibiting mutationally activated enzymes, such as kinases, that are required to maintain cancer cell proliferation and survival. However, in solid tumors the role of the ABL kinases in tumorigenesis is relatively unexplored. ABL1 has been implicated to be activated and promote phenotypes associated with cancer, by mechanisms including loss of the ABL1 negative regulator FUS1 (Lin et al, 2007). ABL kinases regulate a range of cellular processes including cell survival, proliferation, motility, cell polarity and oxidative stress responses, suggesting deregulation of the kinases may promote tumorigenic phenotypes (Greuber et al, 2013). However, the exact role of genetically altered ABL kinases in solid tumors remains largely unknown (Greuber et al, 2013). Depending on cellular context and cancer type, the proliferative effects of ABL1 can be exerted by controlling upregulation of c-myc or cyclin D1 (Furstoss et al, 2002; Kundu et al, 2015). In addition, in breast cancer, ABL1 has been reported to phosphorylate the sliding clamp protein, PCNA at Tyr211 leading to association with chromatin and increased cell proliferation (Zhao et al, 2014). Lastly, both ABL1 and ABL2 are essential for the invasive phenotype of melanoma cells, where ABL kinases were shown to directly regulate expression and activation of matrix metalloproteinases (Ganguly et al, 2012). Combined this highlights that deregulation of the ABL kinases will contribute to acquisition of tumorigenic phenotypes in solid tumors.”

3. What is the mutational profile of the cell lines analyzed? Are other kinases such as SRC amplified in those cell lines?

Response: We thank the reviewer for this comment and have now added mutational profiles for all cell lines analyzed (data set 1). We have also included CNA (copy number alteration) data for each cell line for every gene (data set 1). There are no amplifications of SRC in these cell lines, but that was an excellent point raised by the reviewer.

4. Is there any relationship between ABL mutations and the histological subtypes of lung cancer?

Response: We have now performed this analysis and in general there is no relationship between histological subtype and ABL mutations. In general ABL1 mutations are observed in NSCLC (adenocarcinoma and squamous cell carcinoma) as well as SCLC at low frequencies ranging from 1-3% (Appendix Fig S1A). We have now included this information in the introduction of our manuscript as follows:

“ABL1 and ABL2 display mutations in approximately 1.5% and 4% of NSCLC cases respectively and ABL2 is amplified in approximately 8% of NSCLC cases (Appendix Fig S1 A and B, adapted from cBio Cancer Genomics Portal). Additionally, ABL1 and ABL2 are mutated at a frequency of approximately 3% or 10% in SCLC, respectively (Appendix Fig S1 A and B).”

5. Have authors evaluated if there is CNV of these genes? It seems that this mechanism may be important in the case of ABL2. The overexpression of ABL2 in normal cells may help to elucidate if

ABL2 overexpression has a functional role in lung cancer and may serve as a potential drug target. ABL2 knockdown in tumor cells by siRNA would also help to clarify this point.

Response: We thank the reviewer for making this point and we have now assessed CNVs in lung cancer. Indeed the reviewer is correct and amplification of ABL2 is an alteration that occurs between 3 and 8 percent in lung adenocarcinoma patients (Appendix Fig S1B). To assess if amplified ABL2 is promoting viability in lung cancer cells with ABL2 amplifications we depleted ABL2 from cells with ABL2 amplification. Two of three cell lines with ABL2 amplifications (H810, H1048, and H1355) had increased expression of ABL2 (H810 and H1048) as assessed by western blot (Fig EV2C). Treatment with imatinib suppressed viability in the H810 cells, without affecting viability of the H1048 or H1355 cells (Fig EV2D). Consistent with these results, depletion of ABL2 with three unique siRNAs suppressed viability of H810 cells, but did not alter viability of H1048 or H1355 cells (Fig EV3B). Overall these results indicate that amplified ABL2 is not a major survival gene in lung adenocarcinoma. In addition, we were unsuccessful in generating Beas-2B cells where we could induce expression of WT ABL2 and monitor for changes in tumorigenic phenotypes due to technical issues (Reviewer Fig 2). Despite three attempts, ABL2 was either not expressed or was overexpressed even without tetracycline induction. Due to time constraints we were unable to try a fourth time to generate these inducible cells. We feel these additional experiments expand the scope of our manuscript and again thank the reviewer for these insightful comments.

Reviewer Figure 2

6. Authors use a treatment with 40 nM and 123 nM dasatinib. GI50 value determination and dose-response curves would be useful to determine the kinetics of the treatments. The same comments are applicable to imatinib results.

Response: We have now included the dose response curves for both imatinib and dasatinib (Fig EV2 A and B).

7. What is the effect of dasatinib on colony formation? Conversely, what is the effect of imatinib treatment on cell viability? had these compounds any effect on apoptosis?

Response: We have now assessed the effect of dasatinib on colony formation on a panel of the lung adenocarcinoma cell lines. In general more toxicity is observed, which is to be expected since dasatinib is a less specific ABL inhibitor. The H1915 cells with the R351W homozygous activating mutation and A549 cells are the most sensitive in the colony formation assays and these data are consistent with the MTT assays (Fig 1C and Appendix Fig S2B). The H650 and H1623 that harbor mutations in ABL2 are not sensitive to dasatinib treatment in the colony formation assays (Appendix Fig S2B). To assess the effects of imatinib on viability we treated the panel of lung adenocarcinoma cells lines with imatinib and monitored viability via the MTT assay. Consistent

with the colony formation assay, the H1915 and H2110 cells were the most sensitive to imatinib (Fig 1A and Appendix Fig S2A).

We also assessed the mechanism of imatinib sensitivity in H1915 and H2110 cells. For H1915 cells we observed imatinib caused an inhibition of the cell cycle, which correlated with a significant increase of p21 protein levels (Fig EV4A) and an increase in the percentage of cells in G1 phase and a decrease in the percentage of cells in S-phase of the cell cycle (Fig EV4B). For the H2110 cells treated with imatinib, we observed a decrease in p21 expression levels and an increase in cleaved caspases 3, 7, and 9 as well as an increase in cleaved PARP (Fig EV4A). This correlated with a significant increase in early and late apoptotic cells (Fig EV4C).

8. What is the effect of the induced expression of mutated forms of ABL1 in the proliferation and cell survival of the Beas-2B cells? Did the expression of these mutated forms induce the malignant phenotype of non-tumor cells? Soft agar colony formation assays would help to elucidate the ability of cellular anchorage-independent growth.

Response: Expression of the mutated forms of ABL1 alone did not induce malignant phenotypes in the Beas-2B cells (Reviewer Fig 3A). However, expression of WT ABL1 resulted in a decrease in viability that was not observed upon expression of the G340L or R351W ABL1 mutants (Reviewer Fig 3A). This is likely due to nuclear exclusion of the mutant forms of ABL1 (Fig 5 A-C), as nuclear WT ABL1 is associated with pro-apoptotic and growth inhibitory phenotypes. It is likely that other mutations would have to be introduced into the Beas-2B cells to induce malignant phenotypes. Additionally, we performed soft agar CFA with these cells, however we were unable to obtain meaningful results since the colonies were too small even after weeks of growth.

9. What is the effect of mutations found in primary tumors on proliferation and survival of non-tumor cells?

Response: Tetracycline inducible expression of the additional mutants found in primary tumors enhanced downstream activation of CRKL in Beas-2B cells, similar to that observed when these mutants were expressed in the H157 cells (Figs 3F and EV5). However, similar to the results discussed above, expression of these cancer mutants did not enhance proliferation or survival of Beas-2B cells (Reviewer Fig 3A and B), but in general did not suppress viability compared to WT ABL1 (Reviewer Fig 3B).

Reviewer Figure 3

Reviewer Figure 3 A and B. Colony formation assay with included representative images of colonies formed for Beas-2B cells with and without tetracycline inducible expression of wild type ABL1 and mutant ABL1 variants found in (A) NSCLC cell lines (with representative western blot analysis of ABL1 expression) or (B) in the primary lung tumors. Error bars represent mean of \pm S.E.M., *P < 0.001.

10. Authors conclude that ABL1 mutations increase cytoplasmic localization. It would be worthy to extend this study to all the panel of cell lines included in this study.

Response: We have now examined ABL1 expression in a panel of the cell lines and nuclear exclusion of ABL1 is specific to the cancer mutants, R351W and G340L (Fig 5 A and B).

Did authors perform specific controls of the immunohistochemical technique?

Response: Yes, we have verified localization and specificity of the antibody by using cells depleted of ABL1 as a control (Appendix Fig S9).

11. Authors have based their "in vivo" results on one ABL1 mutated cell line. These results need to be confirmed with the inclusion of other cell lines with different mutations. The inclusion of a group with the gatekeeper construction ABL1-T315I would help to clarify the results.

Response: We thank the reviewer for the comment and agree with the reviewer and have now expanded our *in vivo* data to include a xenograft mouse model where we induced expression of the drug resistant gatekeeper mutant and this promoted resistance to imatinib, which we feel clarifies our results (Fig 3C).

Minor comments:

The percentage of NSCLC patients in the "Introduction" and "The paper explained" sections should be the same.

Response: We thank the reviewer for pointing this out and we have now corrected this mistake.

Referee #2 (Remarks):

EMM-2015-05456

Testoni et al. "Somatically Mutated ABL1 Represents an Actionable and Essential NSCLC Survival Gene "

The authors study the infrequently mutated genes ABL1 and ABL2 as potential therapeutic targets in non-small cell lung cancer (NSCLC). They find NSCLC cell lines with ABL1 mutations (a few lines) and those with ABL1 mutations are sensitive to the ABL kinase inhibitor imatinib, *in vitro* and *in vivo* (xenograft). Those lines with wild type ABL1 and ABL2 mutations are not sensitive. They provide evidence that NSCLCs lines with ABL1 mutations have activated known down stream targets such as pCRKL. Structural modeling studies of the mutant ABL1 protein are also provided. They conclude mutant ABL1 is a new therapeutic target for NSCLC. No clinical therapeutic trials in patients with mutated ABL1 are presented. Besides pharmacologic data on growth inhibitory effect targeting ABL1, siRNA mediated knockdown studies of ABL1 are presented. However, no "rescue" experiments for these siRNA studies are presented to show the siRNA effect was "on target."

Comments to the authors.

This is a technically well-done study. Despite all of the structural and functional studies, the key issues for this to be clinically important are first the quantitative nature of growth inhibition of ABL1 mutant NSCLCs, and then the nature of any acquired resistance mechanisms. With regard to the quantitative effects, they appear to be quite modest compared to other mutant tyrosine kinase targeted therapy in NSCLCs such as targeting mutant EGFR or targeting BCR-ABL in CML. The hard, cold facts are in Fig 1C with an ~50-60% inhibition of colony formation, not even one log and in Fig 3H, Showing a very modest xenograft effect.

If we knew that even anecdotal patients had clinically significant and beneficial responses, these data would be fine. Alternatively, if they had tested the drug in a patient with mutant ABL1 and there were not such effects, that would be fine also, we would know the answer and see that the modest preclinical studies were actually informative about the lack of clinical effects.

If I were the authors, I would try to do some studies showing effect on things like lung cancer stem cells, test of passage as a xenograft after treatment to demonstrate that tumorigenic potential was significantly reduced, or tests of re-plating of tumor cells from colonies after imatinib treatment to show with these more sensitive tests, there was the potential for a large therapeutic effect of imatinib treatment. In addition, if they had developed an imatinib resistant variant and potentially showed the mechanism of resistance was similar or different than that found in CML it would have been interesting no matter what the result was.

Response: We are pleased the reviewer felt our study was technically well done and we thank for this positive feedback. In addition, we thank the reviewer for their scientific suggestions. We have performed the re-plating experiment and these results indicate that the H1915 lung cancer cells harboring the R351W GOF mutation are more sensitive to imatinib treatment in subsequent treatments and that cells often do not survive the re-plating, indicating there is an increased sensitivity of the cells to imatinib that is able to be more accurately assessed with this approach (Appendix Fig S3A). No mutations, other than R351W (c1051t), in ABL1 were observed in re-plated cells indicating they had not yet developed resistance to imatinib (Appendix Fig S3B). However inducible expression of the drug resistant mutant (T315I) in H1915 cells does promote resistance indicating that cells that acquire this mutation will develop resistance to imatinib (Fig 2 A-C). We also recognize that the responses are not complete and have emphasized that imatinib would likely need to be used in combination with other chemotherapies or targeted therapies throughout the manuscript. Unfortunately because the mutations are somewhat rare (approximately 1.5%) there have not been any efforts to our knowledge to treat lung cancer patients with an ABL1 mutation with imatinib. However, we are pushing forward an effort to identify lung cancer patients with ABL1 mutations as part of the TARGET trial at the Christie hospital in Manchester to stratify these patients for treatment with imatinib and concomitantly develop a PDX model and hope to contribute to assessing this in the future.

Referee #3 (Comments on Novelty/Model System):

The work could benefit from some quantitative biochemistry to test the impact of the NSCLC mutations on the catalytic properties of Abl kinase activity.

Referee #3 (Remarks):

This work characterizes several novel mutations in Abl1 in NSCLC cells lines and identifies them as possible targets for therapeutic intervention in cancer. The work is for the most part rigorous and well done.

The only major limitation of this work stems from the lack of comprehensive characterization of impact of the Abl1 mutations on the biochemical activity of Abl. This analysis was limited to assessing the impact of mutant Abl1 overexpression on levels of CrkL phosphorylation and to assessing the relative partitioning of the Abl1 mutants to cytoplasm vs. nucleus. There was some speculation on the potential effects of the mutations on conformational dynamics, but this was based entirely on molecular dynamics simulation. A much more straightforward and direct method to address the impact of these mutations would be to express them in cells and measure the k_{cat} and K_m of the mutants for CrkL in vitro. This would indicate whether and how the mutations directly impact kinase activity.

Response: We thank the review for the positive nature of his comments. We agree that an *in vitro* kinase assay would be informative, however we struggled to IP equal amounts of protein from cells as the WT ABL1 seemed to be more stable in IPs. Interestingly even with significantly lower levels of the R351W immunoprecipitated from cells, this mutant showed similar levels of activity to WT ABL1 consistent with this mutation conferring an increase in activity for the ABL1 kinase (Reviewer Fig 4). Given the scope of the revisions we have attempted to address we were unable to optimize conditions for the assay within the given timeframe.

Reviewer Figure 4

Reviewer Figure 4 *In vitro* kinase assay for mutant ABL1 activity relative to the wild type kinase. HEK293T cells were transiently transfected with HA-tagged wild type ABL1 kinase (WT) or the ABL1 mutants (R351W and G340L) or empty vector control (EV). 48h later the ABL1 kinases were immunoprecipitated and subjected to kinase assay with recombinant GST tagged CRKL substrate. Positive control included kinase assay with the recombinant GST tagged ABL1 kinase. The negative control included GST-CRKL substrate only.

4th Editorial Decision

17 November 2015

Thank you for the submission of your revised manuscript to EMBO Molecular Medicine. We have now received the enclosed reports from the referees that were asked to re-assess it. As you will see the manuscript passes with flying colours and I am thus pleased to inform you that we will be able to accept your manuscript pending the following final amendments:

- 1) As per our Author Guidelines, the description of all reported data that includes statistical testing (including supplemental) must state the name of the statistical test used to generate error bars and P values, the number (n) of independent experiments underlying each data point (not replicate measures of one sample), and the actual P value for each test (not merely 'significant' or 'P < 0.05').
- 2) The excel dataset needs to become an EV dataset, which implies including a legend in the file and adjusting callouts in the manuscript accordingly.
- 3) Please reference the Appendix material and methods section in the main manuscript.
- 4) Please correct the "Appendix Table S3" reference in the main manuscript (should be Appendix Table S2?). Also, please use correct spelling for "appendix".
- 5) We are now encouraging the publication of source data, particularly for electrophoretic gels and blots, with the aim of making primary data more accessible and transparent to the reader. Would you be willing to provide a PDF file per figure that contains the original, uncropped and unprocessed scans of all or at least the key gels used in the manuscript? The PDF files should be labeled with the appropriate figure/panel number, and should have molecular weight markers; further annotation may be useful but is not essential. The PDF files will be published online with the article as supplementary "Source Data" files.

If you have any questions regarding any of the above, just contact us.

I have slightly edited the title and the "The Paper Explained" section. I am attaching the amended manuscript with track changes. Please modify/accept as you find appropriate and work on this version when resubmitting your manuscript

I have also edited the synopsis (must be third person) as below. Again, please modify/accept as you find appropriate.

Somatically mutated ABL1 is defined as a novel druggable driver in lung adenocarcinoma where mutant forms of ABL1 kinase promote increased survival and proliferation, altered localization, and enhanced downstream pathway activation.

- Lung cancer cells harboring ABL1 mutations are sensitive to ABL inhibitors
- Depletion of mutant ABL1 suppresses lung cancer cell viability
- ABL1 mutations affect localization and increase downstream pathway activation
- ABL1 mutants promote cell survival and proliferation

I look forward to reading a new revised version of your manuscript as soon as possible and in any case no later than two weeks from now.

***** Reviewer's comments *****

Referee #1 (Comments on Novelty/Model System):

The Authors have satisfactorily addressed my concerns and I recommend publication.

Referee #1 (Remarks):

In the revised submission, Testoni et al have adequately addressed the concerns raised in my initial review

Referee #2 (Remarks):

The authors have responded to all of the reviewers' comments.

Referee #3 (Remarks):

The authors have tried to address the concern regarding kinase activity and were unsuccessful. These measurements would entail significant additional investment of time and this should not be a barrier to publication of an otherwise very nice study.

2nd Revision - authors' response

25 November 2015

Thank you again for serving as the editor on our manuscript entitled "Somatically Mutated ABL1 Represents an Actionable and Essential NSCLC Survival Gene". As you can imagine we were extremely pleased with the outcome from this round of review and were pleased to know that we adequately addressed the concerns of each reviewer. We have now made the final amendments that you highlighted in the decision letter as follows:

- 1) We have now included the name of the statistical test used to generate error bars and P values, the number of independent experiments underlying each data point, and the actual P value for each test. Specifically we have added these to the figure legends for each figure.
- 2) We have changed the excel file to an EV dataset and adjusted the callouts in the manuscript to match this change in name.
- 3) We have now included a reference of the Appendix materials and methods at the end of the Material and Methods section in the manuscript.
- 4) We now use the correct spelling for appendix and use the correct reference (Appendix Table S2) in the manuscript.

5) We have included a PDF file that includes the original, uncropped and unprocessed scans of almost every key gel used in the manuscript and expanded view sections.

In addition thank you for taking the time to make edits to our manuscript and the synopsis section to make the manuscript more clear and have synopsis written in third person. We have carefully followed the instructions when resubmitting our manuscript. Please do not hesitate to contact us if you require any additional details or changes. Our interactions with EMBO Molecular Medicine for this manuscript have been professional and positive and we look forward to submitting future manuscripts to EMBO Molecular Medicine.

We hope that the revised manuscript is now suitable for publication in EMBO Molecular Medicine.